# Interaction between Immunotherapy and Antiangiogenic Therapy for Cancer

**DOI:** 10.3390/molecules25173900

**Published:** 2020-08-26

**Authors:** Koichi Furukawa, Tatsuya Nagano, Motoko Tachihara, Masatsugu Yamamoto, Yoshihiro Nishimura

**Affiliations:** Division of Respiratory Medicine, Department of Internal Medicine, Kobe University Graduate School of Medicine,7-5-1 Kusunoki-cho, Chuo-ku, Kobe 650-0017, Japan; kwmtn083@med.kobe-u.ac.jp (K.F.); mt0318@med.kobe-u.ac.jp (M.T.); myamamot@med.kobe-u.ac.jp (M.Y.); nishiy@med.kobe-u.ac.jp (Y.N.)

**Keywords:** immune, checkpoint, inhibitor, programmed cell death-1 (PD-1), PD-1-ligand-1 (PD-L1), vascular endothelial growth factor (VEGF)

## Abstract

Although immunotherapy has led to durable responses in diverse cancers, unfortunately, there has been limited efficacy and clinical response rates due to primary or acquired resistance to immunotherapy. To maximize the potential of immunotherapy, combination therapy with antiangiogenic drugs seems to be promising. Some phase III trials showed superiority for survival with the combination of immunotherapy and antiangiogenic therapy. In this study, we describe a synergistic mechanism of immunotherapy and antiangiogenic therapy and summarize current clinical trials of these combinations.

## 1. Introduction

For many years, traditional cancer treatments include surgery, radiotherapy and chemotherapy [1]. Recently, cancer immunotherapy, particularly immune checkpoint therapy targeting programmed cell death-1 (PD-1), PD-1-ligand-1 (PD-L1), and cytotoxic T-lymphocyte antigen 4 (CTLA-4), has become a new treatment option for various types of cancers [2,3]. Immunotherapy has shown dramatic effects and demonstrated long-term survival benefits in some cancers, such as lung cancer [4] and renal cancer [5]; therefore, it has become the first-line treatment for these advanced cancers. However, immunotherapy is not effective for all cancer patients, and even if it is effective at first, some patients experience therapy resistance; therefore, it is necessary to increase the antitumor effect of immune checkpoint inhibitors and overcome resistance. Based on the above findings, combination therapies with immunotherapy and other treatments, such as antiangiogenic therapy, radiation therapy, chemotherapy and surgery, are attracting attention.

Antiangiogenic therapy is a notable strategy for cancer. Tumor cells need additional oxygen and nutrition from vessels as they grow; hence, it is necessary for tumor cells to induce new vessels from the surrounding vasculature, and this process is called angiogenesis [6]. Angiogenesis is strongly associated with hypoxia and is regulated by various growth factors and receptors, including hypoxia-inducible factor (HIF), vascular endothelial growth factor (VEGF), VEGF receptor (VEGFR), platelet-derived growth factor (PDGF), and fibroblast growth factor-2 (FGF2) [7]. Therefore, it was proposed that the inhibition of these signaling pathways and angiogenesis is effective and promising for cancer treatment, and particularly, the VEGF family has been focused on as a key driver of angiogenesis because it is frequently overexpressed in various cancers [8]. Recently, it has become clear that some kinds of VEGF receptors are expressed in macrophages, lymphocytes and dendritic cells [9], and those VEGF signals may lead to immunosuppression [10]. Therefore, VEGF inhibitors are used not only to prevent angiogenesis and normalize vascular permeability in the tumor microenvironment, [11] but also to promote the differentiation and function of immune cells [12,13,14].

As described above, both of these two treatments are remarkable cancer treatments and have begun to be used together to increase the antitumor effect of immunotherapy. Therefore, we review a mechanism of the combination of immunotherapy and antiangiogenic therapy and summarize current clinical trials of these treatments while focusing on the possibilities of combination therapy.

## 2. Cancer Immunotherapy

Since the beginning of the 21st century, cancer immunotherapy has revolutionized cancer treatments, showing dramatic effects in some cancers. This treatment has evolved from the hypothesis of “immune surveillance”, which is the mechanism by which cancer cells are recognized and killed by host immune systems. However, during the development of cancer cells, they grow through the immune surveillance system and emerge as cancer. For this reason, the new hypothesis of “cancer immune editing” was proposed by Dunn et al., which consists of three phases [15]: the “elimination phase” in which cancer cells are eliminated by host immune systems, the “equilibrium phase” in which the immune system is antagonized with cancer cells that are not eliminated, and the “escape phase” in which cancer cells acquire mechanisms that allow them to escape the immune surveillance system and begin to proliferate [16]. This phenomenon has been termed adaptive immune resistance. Therefore, it is considered that the presence of overt tumors means the state of “cancer immune escape” or “adaptive immune resistance”, where cancers form an immunosuppressive environment and change their phenotypes in response to the attack of host immune cells [17].

The immune surveillance mechanism of cancer cells involves various immune cells, particularly cytotoxic T lymphocytes (CTLs), which have an important role in recognizing various antigens, especially those expressed on cancer cells, and directly eliminating them via the T cell receptor (TCR). However, CTLs also express immune checkpoint molecules to prevent autoimmune disease damage, which can be utilized by cancers to suppress T cells and escape immune surveillance. Although there are several other types of effective cancer immunotherapies, such as oncolytic viruses, cancer vaccines, adoptive cell transfer, monoclonal antibodies, and immune checkpoint inhibitors (ICIs) [18,19,20,21], overcoming this suppression mechanism of T cells is the most important common goal, and ICIs have been investigated with the most interest and are now widely used in clinical practice [22].

Immune checkpoint molecules such as PD-1 [23], CTLA-4 [24], T cell immunoglobulin and mucin domain-3 (TIM-3) [25], and lymphocyte-associated gene 3 (LAG-3) [26] have been found to be expressed in several immune cell types, including T cells, natural killer (NK) cells, B cells, dendritic cells, tumor-associated macrophages (TAMs) and myeloid-derived suppressor cells (MDSCs). Among these immune cells, T cells, especially CTLs, are exhausted, and the anticancer functions of the immune system are weakened by these negative costimulatory molecules binding with ligands of cancers, including PD-L1 and B7 (CD80/CD86), of which ICIs act as negative regulators, resulting in the reactivation of anticancer functions and the infiltration of immune cells into tumors. In 2011, the U.S. The Food and Drug Administration (FDA) approved ipilimumab for the treatment of advanced melanoma, followed by pembrolizumab and nivolumab in 2014. Since then, ICIs have provided durable clinical benefits in diverse cancer patients, including breast, lung, kidney, bladder and prostate cancers, as well as melanoma and lymphoma [27,28]. Currently, there are seven ICIs that can be used in clinical practice: anti-CTLA-4 antibody (ipilimumab), anti-PD-1 antibody (pembrolizumab, nivolumab and cemiplimab), and anti-PD-L1 antibody (atezolizumab, durvalumab and avelumab). Table 1 shows the ICIs that can be used for each type of cancer.

## 3. Tumor Immune Microenvironment (TIME)

Although ICIs have provided durable clinical benefit in diverse cancer patients, unfortunately, limited efficacy and clinical response rates have been achieved in cancer patients due to primary or acquired resistance to ICIs [29]. To overcome this tumor resistance, it is vital to understand the tumor immune microenvironment (TIME), which shows an immunosuppressive effect and plays crucial roles in tumor growth, angiogenesis and metastasis, leading to cancer evasion and resistance to ICIs [30,31,32,33,34]. The TIME refers to the local biological environment around solid tumors, which consists of immune cells, microvessels, lymphatic channels, stromal cells, endothelial cells, extracellular matrix, fibroblasts and some signaling molecules, such as chemokines. Immune cells include T and B lymphocytes, NK cells and TAMs. Among them, immune cells and microvessels are key components of the TIME and are responsible for cancer invasion [35]. 

It has been proposed that the TIME can be classified into four types based on the presence of tumor-infiltrating lymphocytes (TILs) and PD-L1 expression [36]. Type I is the stage of PD-L1 positivity with TILs driving adaptive immune resistance, type II is the state of PD-L1 negativity with no TILs indicating immune ignorance (immunologic ignorance), type III is the state of PD-L1 positivity with no TILs indicating intrinsic induction, and type IV is the state of PD-L1 negativity with TILs indicating the role of other suppressors in promoting immune resistance [2,37]. Among these types, cancers in types I and IV are described as the “T cell-inflamed phenotype” or "hot tumors", in which lymphocyte infiltration is common and leads to enhanced therapeutic effects of ICIs [38]. On the other hand, cancers in the other types are described as the “non-T cell-inflamed phenotype” or "cold tumors", in which there is a lack of lymphocyte infiltration but the presence of other immune populations or myeloid cells, leading to diminished effects of ICIs. Actually, "immunoscore" has recently been proposed as a new concept of tumor immunity to quantify TILs in the tumor center and around the tumor, and it has been reported to have a strong association with the prognosis mainly in colorectal cancer [39,40]. Considering this, it is useful to convert immunologically cold tumors to hot tumors, and the angiogenesis of microvessels in the TIME is associated with this conversion.

The TIME in solid tumors is characterized by hypoxia, acidosis, oxidative stress, high lactate levels and declining nutrient resources. As tumor cells grow, these features become more apparent and lead to tumor heterogeneity and genetic instability, resulting in cancer progression and the development of resistance to anticancer therapies [41]. Among these features, hypoxia is known to be the main factor for tumor progression and resistance to chemotherapy and radiotherapy in nearly all solid tumors [42,43,44]. Moreover, it has been revealed that hypoxia contributes to angiogenesis in the TIME and resistance to immunotherapy [45,46,47].

The rapid growth of tumor cells requires a large amount of oxygen that cannot be supplied from the surrounding blood, resulting in hypoxia, which is the imbalance between increased oxygen consumption and inadequate oxygen supply [48]. Persistent hypoxia stimulates the growth of new vasculatures to adapt to low levels of oxygen and nutrients, but these vasculatures are unorganized and have an irregular distribution, contributing to an increased distance between the capillaries, loss of capacity to diffuse oxygen and intratumoral oxygen gradients [49,50]. These vasculatures are also leaky and cause vascular hyperpermeability, resulting in increased interstitial fluid pressure and reduced drug penetration in the TIME because their endothelial structures are discontinuous and lymph vessels are obstructed [51,52].

In this response triggered by hypoxia, hypoxia-inducible factor 1 (HIF-1) plays a key role. HIF-1 is a dimeric protein composed of two subunits expressed in the cells, the oxygen-sensitive α-subunit, HIF-1α, and a constitutively expressed β-subunit, HIF-1β [53,54]. At normal oxygen concentrations, prolyl hydroxylase domain-2 (PHD-2), which is the HIF proline hydroxylase enzyme, generates hydroxyl groups from oxygen molecules and adds them to the Pro402 and Pro564 residues of HIF-1α. Subsequently, von Hippel-Lindau (VHL) E3 ubiquitin ligase binds to HIF-1α, whose proline residue is hydroxylated, resulting in rapid degradation. Under hypoxic conditions, however, the activity of PHD-2 is low, and PHD-2 cannot bind and degrade HIF-1α. Activated HIF-1α translocates to the nucleus, interacts with the HIF-1β subunit, and binds to the promoter regions of various genes that are involved in the hypoxia response, such as angiogenesis, cell proliferation, glucose metabolism and macrophage polarization into TAMs, resulting in the suppression of innate and adaptive antitumor immune responses [55,56,57] (Figure 1). In this HIF1-related signal, VEGF is strongly associated with these genes identified as the transcriptional target of HIF-1, and several studies have shown functional cross-talk among regulatory T cells (Tregs), TAMs, and MDSCs [58]. VEGF is the most potent endothelial-specific mitogen that recruits endothelial cells into hypoxic and avascular areas and stimulates their proliferation [59]. Therefore, VEGF has not only vascular biological effects, including endothelial cell proliferation, induction of vascular permeability and elevation of interstitial fluid pressure but also several immunological effects in the TIME, such as inhibition of dendritic cell maturation resulting in the inactivation of CTLs, recruitment of Tregs, MDSCs and TAMs, the polarization of macrophages from M1 to M2, upregulation of the expression of PD-1 on CD8+ CTLs and Tregs via VEGFR2-dependent signals and induction of Fas ligand expression leading to CTL exhaustion [60,61,62,63,64]. In addition, Tregs themselves produce IL-4, IL-10, and IL-13, inducing monocyte differentiation into TAMs [65] (Figure 2). Because of these immunosuppressive effects of VEGF in the TIME, antiangiogenic therapy targeting VEGF signaling has the possibility to convert “cold” tumors into “hot” tumors with a favorable microenvironment. Additionally, the combination therapy of ICIs and tumor angiogenesis inhibitors seems very reasonable considering the above TIME changes.

## 4. Cancer Immunity Cycle

It was proposed that the repetition and progression of seven stepwise series of events, called the cancer-immunity cycle (CIC), are necessary for the anticancer immune response [66]. In brief, the CIC is composed of cancer antigens (step 1), the presentation of cancer antigens (step 2), priming and activation (step 3, priming phase), T-cell migration (step 4), the tumor infiltration of T cells (step 5), the recognition of tumor cells (step 6), and the destruction of tumor cells (step 7, effector phase). Tumors evade immune surveillance by obstructing one or several steps of the CIC, leading to resistance to immunotherapy and tumor progression. In this cycle, ICIs mainly act at the step where T cells are activated (step 3) and at the step where activated T cells destroy cancer cells (step 7). Here, we focus on the interaction between VEGF signaling and the CIC.

Step 1 refers to the release and capture of tumor neoantigens by dendritic cells (DCs). DCs have major histocompatibility complex (MHC) I and II molecules that present tumor peptide antigens to T cells with costimulation by the B7 molecule, resulting in the efficient activation of T-cell responses against cancer antigens, which is called the priming phase (step 2 to 3). In these steps, tumors release several factors that negatively affect the maturation of DCs [61]. Tumor-derived VEGF is one of these negative factors and is responsible for the differentiation of hematopoietic progenitor cells to DCs, which inhibits the phosphorylation and corresponding degradation of IκB, resulting in the attenuation of NF-κB activation in immature DCs [67,68]. Thus, VEGF inhibits the functional maturation of DCs and promotes immune evasion.

Steps 4 and 5 refer to the migration and infiltration of primed and activated T cells from the lymph node to the tumor. Tumor blood vessels promoted by VEGF can inhibit the extravasation of immune cells from blood to the intraepithelial spaces by downregulating the expression of the adhesion molecules required for the adhesion and migration of lymphocytes, such as intercellular adhesion molecule (ICAM) and vascular cell adhesion molecule (VCAM) [69,70].

Steps 6 and 7 refer to the recognition and destruction of tumor cells, which is called the effector phase. In this phase, VEGF binding to VEGFR promotes immune evasion via the recruitment and expansion of MDSCs through the phosphorylation and activation of the STAT3 signaling pathway [71,72]. MDSCs are heterogeneous populations of cells derived from the bone marrow and consist of polymorphonuclear MDSCs (PMN-MDSCs) and monocytic MDSCs (M-MDSCs) [73]. MDSCs accumulate through persistent inflammation in the TIME, providing signals for their expansion and activation. Activated MDSCs express high levels of PD-L1 and B7-1/2, reduce amino acids such as tryptophan, arginine and cysteine required for T cell activation, induce Treg infiltration into the TIME, and secrete various inflammatory factors, including IL-10, TGF-β and PGE2, resulting in the suppression of antigen-specific T-cell proliferation [74,75]. Additionally, the TIME inhibits the differentiation of MDSCs into mature myeloid cells, such as DCs, mature neutrophils and macrophages, which can promote antitumor immune activities [74]. As these mechanisms occur in each step, VEGF creates an immune suppressive microenvironment and promotes the immune evasion of tumors. Taken together, these findings suggest that the combination therapy of ICI and tumor angiogenesis inhibitors still seems very reasonable and effective.

## 5. Clinical Evidence

The mechanism by which the combination therapy of ICI and anti-angiogenesis therapy exerts a synergistic effect is that anti-angiogenesis therapy not only inhibits angiogenesis but also reprograms TIME [76]. Tumor angiogenesis inhibitors can be broadly classified into two groups: drugs that inhibit the binding of VEGF-A and VEGFR or multikinase inhibitors that are small molecules that inhibit the kinase activity of VEGFR. Many of these inhibitors have been tested in several clinical trials and have shown prolonged survival and progression-free survival [77]. Bevacizumab, which is a recombinant humanized monoclonal antibody targeting VEGF-A, was first approved by the FDA [78]. It blocks the interaction between VEGF-A and VEGFR, primarily VEGFR-1 and VEGFR-2, on the surface of endothelial cells. Bevacizumab is currently approved for the treatment of recurrent, persistent or metastatic cervical cancer, metastatic colorectal cancer, glioblastoma, nonsquamous non-small cell lung cancer (NonSq-NSCLC), ovarian cancer, fallopian tube or primary peritoneal cancer and metastatic renal cell carcinoma (RCC) [79,80,81,82,83]. Similarly, ramucirumab is a monoclonal antibody inhibitor targeting VEGFR2 that is approved for the treatment of advanced gastric cancer or gastroesophageal junction adenocarcinoma, NSCLC, metastatic colorectal cancer, and hepatocellular carcinoma (HCC) [84,85,86,87]. Multikinase inhibitors include sunitinib, axitinib, and sorafenib, which are used for RCC and HCC [88,89]. These angiogenesis inhibitors have been used as monotherapy or combination therapy, mainly with chemotherapy, to enhance the effects of chemotherapy by increasing the intratumor concentration of chemotherapy. Moreover, based on a number of rationales about antiangiogenesis conferring immunosuppressive effects and actual evidence of angiogenesis inhibitors augmenting the benefits of ICIs resulting in durable responses, combination therapy with ICIs has been practiced and used in a variety of cancers.

For example, combination therapies of bevacizumab and atezolizumab, sunitinib or pazopanib and nivolumab, axitinib and pembrolizumab have shown some additional efficacy [90,91,92]. The phase III trials examining combination therapies of ICI and antiangiogenic therapy are summarized in Table 2.

A phase III trial comparing nivolumab/ipilimumab followed by nivolumab with nivolumab/ipilimumab followed by sunitinib in patients with previously untreated RCC (CheckMate 214) showed that nivolumab/ipilimumab followed by nivolumab significantly prolonged overall survival (OS) compared to sunitinib (hazard ratio [HR], 0.66; 95% confidence interval [CI], 0.54 to 0.80; *p* < 0.0001; not reached vs. 26.6 months) [5]. The most common grade 3-4 treatment-related adverse events (AEs) determined by the Common Terminology Criteria for Adverse Events (CTCAE) in the nivolumab/ipilimumab followed by nivolumab group were increased lipase (10%), increased amylase (6%) and increased alanine aminotransferase (5%), whereas in the sunitinib group, they were hypertension (17%), fatigue (10%) and palmar-plantar erythrodysesthesia (9%). On the other hand, another phase III trial comparing atezolizumab/bevacizumab with sunitinib in patients with previously untreated metastatic RCC (IMmotion151 trial) showed that atezolizumab/bevacizumab significantly prolonged PFS compared to sunitinib (HR, 0.74; 95% CI, 0.57 to 0.96; *p* = 0.0217; 11.2 months vs. 7.7 months) [93]. Grade 3-4 AEs were observed in 40% of the atezolizumab/bevacizumab group and in 54% of the sunitinib group. Furthermore, a phase III trial comparing avelumab/axitinib with sunitinib in patients with previously untreated advanced RCC (JAVELIN Renal 101 trial) revealed that avelumab/axitinib significantly prolonged PFS compared to sunitinib (HR, 0.69; 95% CI, 0.56 to 0.84; *p* < 0.001; 13.8 months vs. 8.4 months) [94]. Grade 3-5 AEs were observed in 71.2% of the avelumab/axitinib group and in 71.5% of the sunitinib group. A phase III trial comparing atezolizumab/bevacizumab plus carboplatin/paclitaxel (ABCP) with atezolizumab plus carboplatin/paclitaxel (ACP) or bevacizumab plus carboplatin/paclitaxel (BCP) was conducted in chemotherapy-naive patients with nonsquamous non-small-cell lung cancer (NSCLC) (IMpower150 trial) [95]. In the intention-to-treat population, ABCP showed superior OS compared to BCP (HR, 0.76; 95% CI, 0.63 to 0.93; 19.8 months vs. 14.9 months). Grade 3–4 AEs occurred in 57% of the ABCP group, in 43% of the ACP group and in 49% of the BCP group.

## 6. Conclusions

Anti-angiogenesis therapy not only inhibits the blood vessels that feed tumors but also reprograms the TIME. Some clinical trials that investigated the synergistic effect of the combination therapy acquired promising outcomes in several cancers. This ideal combination may further improve the patient's prognosis.

## Figures and Tables

**Figure 1 molecules-25-03900-f001:**
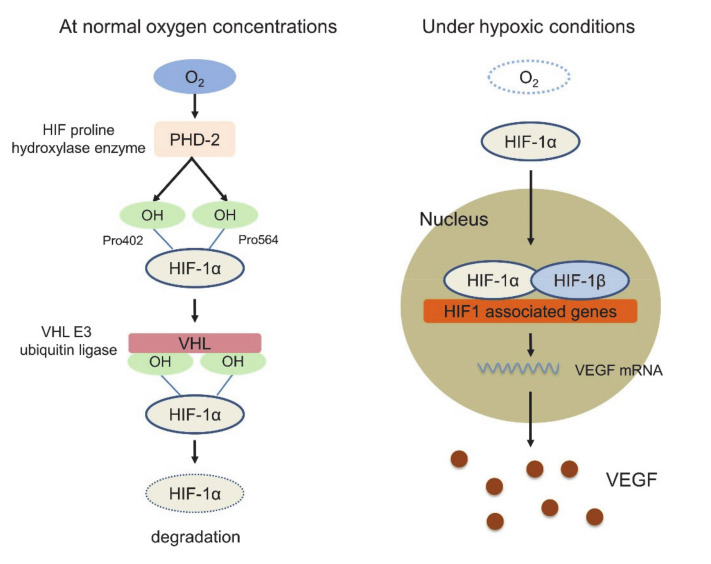
hypoxia-inducible factor 1 (HIF-1) pathway at the normal oxygen concentrations and under hypoxic conditions.

**Figure 2 molecules-25-03900-f002:**
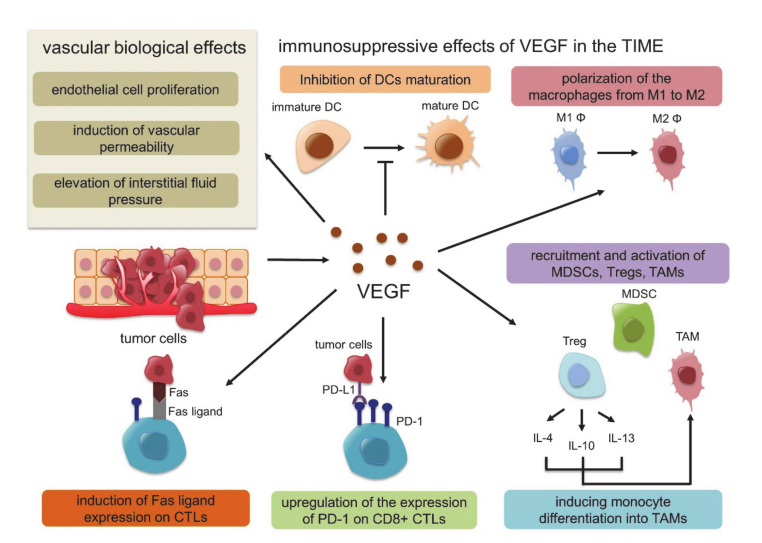
Vascular biological effects and immunosuppressive effects of vascular endothelial growth factor (VEGF) in the tumor immune microenvironment (TIME).

**Table 1 molecules-25-03900-t001:** The Food and Drug Administration (FDA) -approved target cancers of immune checkpoint inhibitors.

ICIs	Target Molecules	Target Cancers
Ipilimumab	CTLA-4	Colorectal cancer, HCC, Melanoma, NSCLC, RCC
Pembrolizumab	PD-1	Cervical cancer, Classic Hodgkin lymphoma, Cutaneous squamous cell carcinoma, Endometrial carcinoma, Gastric cancer, Gastroesophageal junction cancer, HCC, Melanoma, Merkel cell carcinoma, Microsatellite instability-high cancer, Mismatch repair deficient cancer, NSCLC, Primary mediastinal large B-cell lymphoma, RCC, SCLC, Solid tumors, Squamous cell carcinoma of the esophagus, Squamous cell carcinoma of the head and neck, Urothelial carcinoma
Nivolumab	PD-1	Classic Hodgkin lymphoma, Colorectal cancer, HCC, Melanoma, NSCLC, RCC, SCLC, Squamous cell carcinoma of the esophagus, Squamous cell carcinoma of the head and neck, Urothelial carcinoma
Cemiplimab	PD-1	Cutaneous squamous cell carcinoma
Atezolizumab	PD-L1	Breast cancer, HCC, NSCLC, SCLC, Urothelial carcinoma
Durvalumab	PD-L1	NSCLC, SCLC, Urothelial carcinoma
Avelumab	PD-L1	Merkel cell carcinoma, RCC, Urothelial carcinoma

Abbreviations: FDA, Food and Drug Administration; HCC, hepatocellular carcinoma; ICI, Immune checkpoint inhibitor; NSCLC, non-small cell lung cancer; RCC, renal cell carcinoma; SCC, squamous cell carcinoma; SCLC, small cell lung cancer.

**Table 2 molecules-25-03900-t002:** Phase III trials examining combination therapies of immune checkpoint inhibitors (ICIs) and antiangiogenic therapy.

Trial	Cases	Disease	Regimen	RR (%)	PFS (months)	HR	OS (months)	HR
CheckMate 214	1096	RCC	N/I continued vs. Sunitinib	42 vs. 29	8.2 vs. 8.3	0.77 (0.65–0.90) *p* = 0.0014	NR vs. 26.6	0.66 (0.54–0.80) *p* < 0.0001
IMmotion151	915	RCC	A/B vs. Sunitinib	43 vs. 37	11.2 vs. 7.7	0.74 (0.57–0.96) *p* = 0.0217	34.0 vs. 32.7	0.84 (0.62–1.15) *p* = 0.2857
JAVELIN Renal 101 trial	886	RCC	Avelumab/Axitinib vs. Sunitinib	51.4 vs. 25.7	13.8 vs. 8.4	0.69 (0.56–0.84) *p* < 0.001	Immature	Immature
IMpower150	1191	Non-Sq NSCLC	A/B/C/P or A/C/P vs. B/C/P	56 vs. 4041 vs. 40	8.4 vs. 6.8 6.7 vs. 6.8	0.59 (0.50–0.69) 0.91 (0.78–1.06)	19.8 vs. 14.9 19.5 vs. 14.9	0.76 (0.63–0.93) 0.85 (0.71–1.03)

Abbreviations: RR, response rate; PFS, progression-free survival; HR, hazard ratio; OS, overall survival; RCC, renal cell carcinoma; NR, not reached; Non-Sq NSCLC, nonsquamous non-small-cell lung cancer; N, nivolumab; I, ipilimumab; A, atezolizumab; B, bevacizumab; C, carboplatin; P, paclitaxel.

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
