# Peer review of "Interaction between Immunotherapy and Antiangiogenic Therapy for Cancer"

_molecules, 2020, doi:10.3390/molecules25173900_

Round 1

Reviewer 1 Report

In this paper, Koichi Furukawa et al., made a review related with the Interaction between immunotherapy and antiangiogenic therapy for cancer.

The subject of the article are actual. However the synergistic mechanisms referred in the abstract by the auhtors is not well described in the text. They need to clarify this aspect. Further, they need to describe and clarify also the role of immunoscore.

Furthermore, the conclusion need to be improved

Other comments:

  1. The text need to be illustrated with same figures (for instance related with TIME and cancer immunity cycle) and the table 1 aspect needs to be modified in order to be easy to read and more appealing.

Reviewer 2 Report

The review describes the molecular and clinical aspects of the immunotherapy (i.e. immune checkpoint inhibitors - ICIs) and antiangiogenic therapy. Chapter # 3 illustrates an important regulatory role of the tumor microenviroment in the tumor response to ICIs. To my opinion, it's very important and provides the molecular basis for the rationale to use the ocombination therapy of ICIs and antiangiogenic drugs for cancer patients.

In general, the review is very well-wrtten, the language and style are also very good. 

Minor: 

Just a suggestion - given the interaction between VEGF and immune system is complex and includes the multiple mechanisms (e.g.  upregulation of PD-1 on CD8 lymphocytes, maturation of DCs, activity of Tregs, polarization of the macrophages from M1 to M2, etc.), it will be great to include the figure in the manuscript to illustrate these interactions. This practics is very common, especially for the reviews. This figure will be very helpful and illustrative and convince a broad audience that inhibition of VEGF-signaling will have a strong impact on the effectiveness of immunotherapy for the patients with solid tumors.  
